# Unveiling the Dot-Perspective Task: Integrating Implicit-Mentalistic with Sub-Mentalistic Processes

**Cong Fan** [1,2,3,*] [ID], **Tirta Susilo** [3] **and Jason Low** [3]

1 Research Center of Brain and Cognitive Neuroscience, Liaoning Normal University, Dalian 116029, China
2 Key Laboratory of Brain and Cognitive Neuroscience, Liaoning Province, Dalian 116029, China
3 School of Psychology, Victoria University of Wellington, Wellington 6140, New Zealand
* Correspondence: congfan1199@lnnu.edu.cn

**Abstract:** Adults' performances on the dot-perspective task showed a consistency effect: participants were slower to judge their own visual perspective when their own perspective and others' perspective were different compared to when both perspectives were the same. This effect has been explained by two competing accounts: the implicit mentalising account suggests the effect arises from relatively automatic tracking of others' visual perspectives, whereas the submentalising account suggests the effect arises from domain-general attentional orienting. We conducted three experiments to adjudicate between the two competing accounts. Experiment 1 manipulated eye–head directional cues (gaze-averted-face versus head-averted-face) and measured its effect on implicit mentalising (in the dot-perspective task) and attentional orienting (in the Posner task). Eye–head directional cues modulated attentional orienting but not implicit mentalising, supporting the importance of visual access and the existence of implicit mentalising in the dot-perspective task. Experiment 2 compared the effect of gaze-averted versus finger-pointing agents. Finger-pointing direction might induce attentional orienting effects on both tasks. Experiment 3 combined finger-pointing with manipulation of the agent's visual access (eyes-sighted versus eyes-covered) on the dot-perspective task. Visual access did not modulate the consistency effect when finger-pointing was simultaneously displayed. The findings of Experiments 2 and 3 indicated the contribution of the sub-mentalistic process to the dot-perspective task. Overall, the findings suggest that implicit mentalising and submentalising may co-exist in human social perceptual processes. Visual access appears to play a dominant role in modulating implicit mentalising on the dot-perspective task, but the process may be interfered with by finger-pointing cues (more salient than gaze cues) via a sub-mentalistic attentional-orienting mechanism.

**Keywords:** consistency effect; implicit mentalising; submentalising; attentional orienting; cue-validity effect



## 1. Introduction

The capacity to understand a visual scenario from one's own viewpoint and someone else's viewpoint plays a vital role in human social interaction. A core aspect of this capacity is the ability to track what someone else can and cannot see, known as level-1 visual perspective-taking (L1VPT, [1,2]). Two accounts have been proposed to explain what drives L1VPT. The implicit mentalising account claims that L1VPT results from a fast and relatively automatic tracking of an agent's perspective of a visual scene [1]. By contrast, the submentalising account argues that L1VPT arises from attentional orienting driven by the agent's directional cues, such as eye, head, or body directions [3]. We attempted to shed light on these accounts by providing new evidence of the integration of them.

The implicit mentalising account originates from the results of a dot-perspective task designed by Samson and colleagues [1]. The researchers developed a visual scene with a virtual avatar facing the left-side or right-side wall of a room that features discs horizontally displayed on the wall(s). The participants saw a word followed by a number (i.e., 'Self/She'

perspective and the numbers 0/1/2/3) and then were required to make judgements on whether the digit matched the number of dots seen from the participant's own perspective or the avatar's perspective in the subsequently presented visual scene. The intriguing finding was the consistency effect on 'Self-perspective' trials of the task. Participants judged their own visual perspectives more slowly when participants and the avatar saw a different number of discs (i.e., inconsistent condition) relative to when they saw the same number of discs (i.e., consistent condition). The implicit mentalising account takes this effect as a demonstration that adults' own visual perspective can be interfered with by automatically ascribing a seeing mental state through the avatar's eyes even when the avatar's perspective was task-irrelevant (i.e., altercentric interference effect). However, Heyes [3] challenged that L1VPT processing can be attributed to attentional orienting. Specifically, the avatar's head and/or torso directions orient the participants' attention to one lateral wall, which may lead to the consistency effect in the dot-perspective task. The interpretation may be supported by the findings that arrows (with directional property) and avatars elicited similar resulting patterns in terms of the consistency effect [4,5]. The debate over the two competing accounts has become a frequent subject of investigation in the past decade.

For submentalising, Santiesteban et al. [4], for example, added new trials where the avatar was replaced with an arrow having similar low-level directional features. Consistency effects of comparable size were found in the avatar and arrow conditions, suggesting that the effects in the dot-perspective task may be triggered by domain-general processes, such as attentional orienting. Even though arrows, similar to avatars, have been found to trigger a consistency effect, there are limitations to such approaches. First, whilst the arrow that Santiesteban et al. created has directional properties, it can be regarded as having agentic characteristics [6] because its height, shape, colour distribution and area were matched to the avatar (also in Experiment 1 of Conway et al.'s study [7]). Second, studies show that human beings may attribute mental states to simple geometric shapes [8]; therefore, rather than being submentalisers, adults may be, within limits, supermentalisers for arrows. Thus, these findings cannot rule out the implicit mentalising account for relatively automatic L1VPT. Additionally, Cole et al. [9] pointed out that simply replicating the classic arrow cueing effect cannot tell us very much, if anything, about why the dot-perspective effect works.

One approach that has been widely used to clarify the ongoing debate is to manipulate the avatar's visual access. The first study to perform the manipulation in the context of mentalising and perspective-taking was Cole et al. [10], which originated from animal behaviour work. This approach has produced mixed findings. Furlanetto et al. [11] manipulated the avatar's visual access using transparent versus opaque goggles. After learning that transparent googles are associated with seeing and opaque googles with non-seeing, participants completed the dot-perspective task. The authors observed a consistency effect in the seeing condition but not the non-seeing condition. They interpreted that this effect was observed based on automatic tracking of the avatar's visual access, supporting the implicit mentalising account. However, Conway et al. [7] failed to replicate Furlanetto et al.'s [11] findings and found consistency effects in both the seeing and non-seeing conditions. Following the submentalising account, these authors interpreted that the consistency effect resulted from participants' attentional shift as underpinned by the avatar's directional property.

However, there may be shortcomings in both studies. First, it can be challenging for participants to fully appreciate visual access being manipulated by the type of goggles worn by the avatar when the eye region occupies only a small part of the computer-rendered avatar. Second, it can also be challenging for participants to keep in mind the manipulation of visual access when seeing the room scenario, and responses to task instructions need to be completed in a very short duration (i.e., 2 s). To address such issues, Fan et al. [12] manipulated the agent's line of sight by employing an easily identifiable barrier (i.e., a black rectangle covering the agent's eyes). All the participants—in an evaluation before the formal experiment—reported that eyes-opened faces could see, whereas eyes-covered faces could

not see. The authors discovered a significant consistency effect in the visible condition but not the invisible condition. Consistent with Furlanetto et al.'s work [11], the findings lend support to the implicit mentalising account. Even though Cole et al. [9] and Cole et al. [10] employed an easily identifiable barrier, limited sample sizes (i.e., 24 participants and 16 participants, respectively) created bias regarding the reliability of the findings in favour of mentalising or the submentalising accounts.

Gardner et al. directly compared attentional orienting with visual perspective-taking to dissociate the two competing accounts [13]. The authors adopted the Posner task (a classic measure of attentional orienting) and the dot-perspective task and manipulated the avatar stance to ascertain whether the avatar stance would modulate attentional orienting but not visual perspective-taking. Avatar stance was considered to be a directional cue on the basis of a stronger orienting effect for stance-averted avatars (i.e., the avatar's head was directed to one lateral wall, whereas its torso was frontally displayed) than for stance-maintained avatars (i.e., the avatar's head and torso were directed to the same lateral wall) [14]. In Experiment 1, they used the Posner task to document a cue-validity effect; namely, participants detected the targets more slowly at the non-cued locations relative to cued locations. More importantly, attentional orienting was affected by avatar stance, which was reflected by an apparent cue-validity effect for stance-averted avatars but not for stance-maintained avatars. However, the consistency effect in the dot-perspective task of Experiment 2 was not modulated by the avatar stance. The dissociation between attentional orienting and visual perspective-taking supported the implicit mentalising account but cast doubt on the submentalising account.

Nevertheless, in Gardner et al.'s [13] study, the virtual avatars showed clear head and torso directions but unclear information about other facial features, especially eye gaze. As eye gaze is a window into another person's mental state [15], lack of eye information for the avatars may limit the generalisability of Gardner et al.'s [13] findings. Moreover, Wiese et al. [16] proposed that effects related to mentalising generation may be clearer if the agent (i.e., activation stimulus) possesses the property of having a mind. Therefore, pictures of a real person with more mentalising representations (i.e., obvious facial features like eye direction) appear to be more ideal materials to examine L1VPT processing compared to the virtual avatar.

Additionally, it remains important to determine whether a face-related directional cue could distinguish attentional orienting from implicit mentalising to support the existence of implicit mentalising, as the avatar stance did in Gardner et al.'s [13] study. Qian et al. [17] separated eye direction from head orientation by manipulating eye–head directional cues: gaze-aversion face (i.e., a frontal-view face with averted eye gaze) vs. head-aversion face (i.e., lateral-view head with the gazer's eyes looking back to the observer), which are coined based on the mutual relationship between eye direction and head orientation. The researchers confirmed that participants evaluated the gaze direction of gaze-averted faces to be significantly higher than that of head-averted faces. In the formal experiment, for the gaze-cueing task (a modified Posner task), participants showed a stronger cue-validity effect for gaze-averted faces than for head-averted faces. The researchers explained that the head-averted face, a weaker orienting cue, triggered a weaker gaze-cueing effect. The findings demonstrated that gaze-cueing attentional orienting was modulated by perceived gaze direction with reference to head orientation in the modified Posner task.

For the dot-perspective task, if the submentalising account matters, the modulation effect in the Posner task would be generalised to the dot-perspective task via an attentional-orienting mechanism. If the implicit mentalising account involving the role of the agent's visual access matters, perceived gaze direction with reference to head orientation would not modulate implicit mentalising as visual access is always available in both kinds of faces. Then, the potential modulation effect in the Posner task but not the dot-perspective task could dissociate attentional orienting from implicit mentalising. Experiment 1 attempted to shed light on the implicit mentalising versus submentalising debate by first measuring

eye–head cues' (i.e., gaze-averted and head-averted faces) effects in both the Posner task and the dot-perspective task.

In addition to eye and head cues, manipulation of cues from other parts of the human body may also provide insights into clarifying the implicit mentalising vs. submentalising debate. Kendon [18] and McNeill [19] claimed that gestures like finger-pointing play a pivotal role in social communication. Additionally, finger-pointing may also be a more accurate spatial cue than averted gaze for redirecting visual attention to a target [20,21]. One natural question is whether eye–finger-pointing cues (i.e., gaze-averted agent vs. finger-pointing agent) can dissociate attentional orienting from implicit mentalising to clarify the processing mechanism of L1VPT.

In terms of eye–finger-pointing cues, only two studies directly compared the role of gaze direction with that of finger-pointing direction on attentional orienting. Doherty and Anderson [22] reported the superiority of perceiving finger-pointing relative to eye gaze as a directional cue in preschool children. The authors thought that children appeared to have difficulty in reliably judging directional information specific to eye direction. Likewise, Gregory et al. [23] also found an advantage of finger-pointing compared to eye gaze for young children but in an eye-tracking task. Although the cues were non-predictive, finger-pointing triggered a stronger cue-validity effect (i.e., slower eye movement responses to invalid cues compared with valid ones) than eye-gaze did in 3–5-year-old children. One possible explanation was that young children learned the adult's hand cues' connection with the targets earlier than other cues, such as eye gaze cues because hand gestures were more salient. Based on the superiority, we wondered if eye–finger-pointing cues could be a factor in modulating the attentional orienting of adults in the Posner task.

Additionally, for the dot-perspective task, if the submentalising account is involved, the modulation effect induced by eye–finger-pointing cues in the Posner task would be generalised to the dot-perspective task through an attentional-orienting mechanism. If the implicit mentalising account is involved, finger-pointing agents will generate attentional orienting, whereas averted-gaze agents would generate implicit mentalising as visual access was available in averted-gaze agents but not finger-pointing agents. Experiment 2 tried to elucidate the debate by first measuring eye–finger-pointing cues' effects in both the Posner task and the dot-perspective task. It is also critical to check whether visual access could still predominate the processes in the dot-perspective task when finger-pointing was considered. Accordingly, Experiment 3 explored whether the combination of line-of-sight manipulation with finger-pointing can provide new insights into the debate.

Overall, the ongoing debate on whether the consistency effect in the dot-perspective task arises from specific implicit mentalising or domain-general attentional orienting processes appeals to further investigations. In the current study, we sought to open new avenues for clarifying the debate by measuring effects in both the Posner task and the dot-perspective task triggered by the agent's different body parts.

## 2. Experiment 1

In this experiment, we examined whether manipulating eye–head direction cues (i.e., gaze-averted face vs. head-averted face) could modulate attentional orienting (in the Posner task) but not implicit mentalising (in the dot-perspective task). The real human's eye-head directional cue manipulation was based on Qian et al.'s [17] work, revealing that gaze-averted faces induced a stronger cue-validity effect than head-averted faces in a modified Posner task. The findings demonstrated that the gaze-cueing orienting effect was modulated by gaze perception with reference to head orientation in the modified Posner task, which led to a possibility that the eye–head directional cue's manipulation might modulate the orienting effect in the Posner task. For the dot-perspective task, the submentalising account would predict the resulting patterns similar to those of the Posner task. In contrast, the implicit mentalising account emphasising the role of visual access would posit another possibility: the consistency effect might not be influenced by manipulations of the agent's directional property due to visible eyes for both gaze-

averted and head-averted faces. Under this circumstance, attentional orienting in the Posner task could be distinguished from implicit mentalising in the dot-perspective task. Accordingly, Experiment 1 used the Posner and dot-perspective tasks and manipulated the real human's eye–head directional cue, attempting to dissociate attentional orienting from implicit mentalising.

### 2.1. Experiment 1A (Posner Task)

Experiment 1A evaluated whether eye–head directional cue manipulation (i.e., gaze-averted face vs. head-averted face) would modulate attentional orienting via the cue-validity effect in the Posner task. In order to separate gaze direction from head orientation, the gaze angle of head-averted faces was made to be direct to a maximum extent. Following Qian et al.'s [17] work, we predicted that compared with head-averted faces, gaze-averted faces would induce a stronger cue-validity effect.

#### 2.1.1. Participants

Thirty-nine undergraduates were recruited through the Introduction to Psychology Research Programme (IPRP) system of Victoria University of Wellington (VUW) and obtained 0.5-course credits for participation. Thirty-three participants (25 females, mean age: 19.3 years; age range: 18–27 years) remained for further analysis of the formal experiment after excluding six participants (see 'Results'). A priori power analysis conducted using an R package SIMR indicated that the sample size of 27 allowed for the examination of the Validity × Avatar-stance interaction effect found by Gardner et al. [13] at the power of 84.3%. The sample size of 33 exceeded the required sample size. The study was approved by the School of Psychology Human Ethics Committee under the delegated authority of VUW's Human Ethics Committee, which was carried out in line with the Declaration of Helsinki. All participants of the current study reported normal or corrected-to-normal vision and were right-handed. Every participant was given informed consent before participation and debriefed after participation.

#### 2.1.2. Stimuli

We took photographs of a volunteered female undergraduate from VUW (the volunteer signed the informed consent to allow her pictures to be taken and employed for the study). Gaze-averted faces (i.e., a frontal-view head with 45° left- or right-averted eyes) and head-averted faces (i.e., a half-profile head with eyes looking at the observer) were included. Sitting at a distance of about 72 cm from a 14-inch monitor, the participant could observe the scene in a room (13.0° × 12.0°) where a gaze-averted face (3.5° × 4.6°) or a head-averted face (3.5° × 4.6°) with the same width of eyes (0.2°) was centrally presented. The room consisted of left, back, and right walls with one disc (about 0.7° × 0.9°, 3.1° from the face) displayed on one of the lateral-view walls (see examples in Figure 1A,B). The faces were equivalently oriented to the left and right walls. On 50% of trials, the side where the cues were directed was effective for detecting the disc (valid condition) but ineffective on the remaining trials (invalid condition) (see supplemental Appendix A for evaluations of the stimuli).

#### 2.1.3. Procedure

Participants completed the online study with OpenSesame 3.2.5 software (Psychology Software Tools) and the JATOS server. Participants were instructed to complete the target-detection task shown in Figure 2A. Trials commenced with a fixation cross appearing for 750 ms in the centre of the virtual room, followed by a 500 ms interval. Then, a face was centrally presented in the same room with a variable delay (stimulus onset asynchrony, SOA = 100, 300 or 700 ms) (SOAs were considered to measure that the attentional-orienting effect was either reflexive (i.e., 100, 300 ms) or volitional (i.e., 700 ms)). Afterwards, a target (dot) was displayed on one lateral wall for at most 3000 ms, and the participants executed

their responses by pressing the letter "h" as quickly and accurately as they detected the dot. Finally, an inter-trial 500 ms interval was presented with the room being presented.

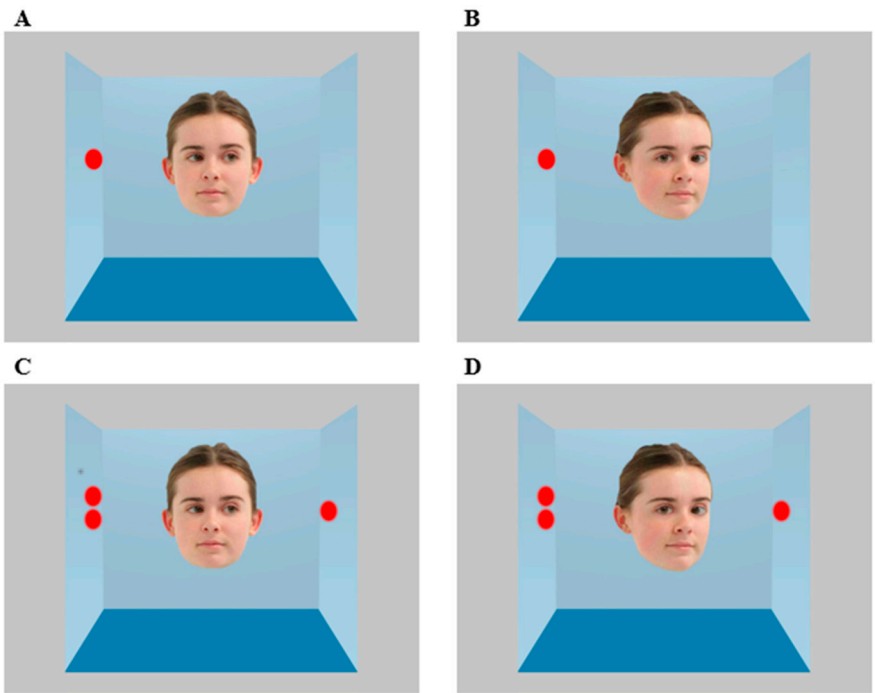

**Figure 1.** Examples of room images used in Experiment 1A ((**A**): Gaze-averted Face, (**B**): Head-averted Face), and Experiment 1B ((**C**): Gaze-averted Face, (**D**): Head-averted Face).

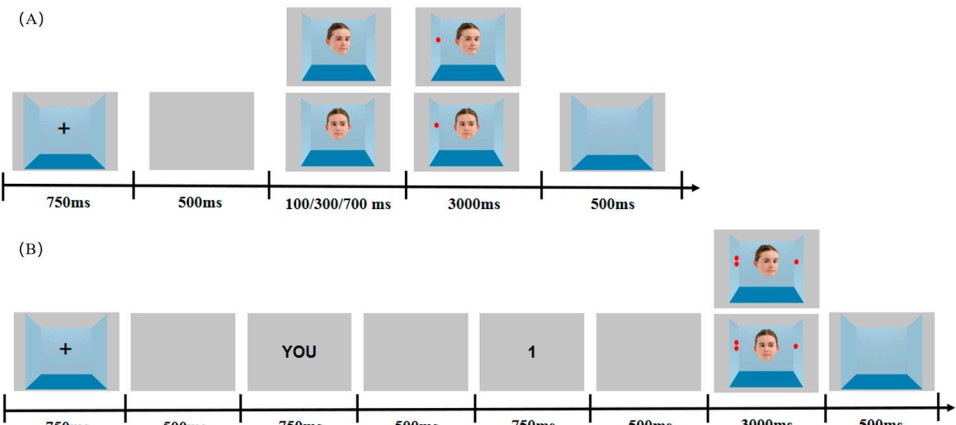

**Figure 2.** Overviews of a representative test trial of Experiment 1A (**A**) and Experiment 1B (**B**).

Following a practice session of 40 trials, the participants completed 4 experimental blocks, with the presentation order of the trials being pseudo-randomised and fixed across participants. Each block's 80 trials contained 24 trials (12 valid trials with gaze-averted face/head-averted face, 12 invalid trials with gaze-averted face/head-averted face) in every SOA (100, 300, 700 ms), and 8 catch trials. In catch trials, a face centrally appeared in the room with no dot and response. The gaze-averted and head-averted faces were displayed in different blocks, with the presentation order counterbalanced across participants.

### 2.1.4. Results and Discussion

The factors 'SOA' (100 ms vs. 300 ms vs. 700 ms), 'Directional cue' (Gaze aversion vs. Head aversion) and 'Validity' (Valid vs. Invalid) were included in the analysis and formed a 3 × 2 × 2 within-subject design. The percentage of error and response time were the key

dependent variables. Participants whose accuracy was less than 90% or whose response time was not within the range 'mean ± 2.5 standard deviations (SDs)' were eliminated from the data set. Three participants with low accuracy and another three participants whose response times were higher than 2.5 standard deviations of mean RT were excluded, leaving data from 33 participants for further analysis. On average, the error rate of catch trials was 3.98%.

In terms of response time, only accurate trials of non-catch trials were analysed. Analysis of variance (ANOVA) showed a significant interaction effect between SOA and Validity ($F$ (2,64) = 5.44, $p$ = 0.007, $\eta_p^2$ = 0.15). Paired-sample t-tests revealed that there was a Validity effect when SOA was 700 ms ($t$(32) = −3.06, $p$ = 0.004) but not 100 ms ($t$(32) = 0.88, $p$ = 0.38) or 300 ms ($t$(32) = 0.023, $p$ = 0.98). Specifically, when SOA was 700 ms, participants were slower to detect the target in the invalid condition (mean ± standard error = 376.37 ± 7.87 ms) when compared to the valid condition (365.68 ± 8.39 ms). In contrast, participants made judgements on target position as quickly in the valid condition as in the invalid condition, both when SOA was 100 ms and when SOA was 300 ms (Figure 3A). Additionally, there was a significant Central stimulus type × Validity interaction effect ($F$ (1,32) = 9.61, $p$ = 0.004, $\eta_p^2$ = 0.23). Paired-sample t-tests indicated a Validity effect when a gaze-averted face ($t$(32) = −3.16, $p$ = 0.003) but not a head-averted face ($t$(32) = 1.59, $p$ = 0.12) was centrally presented. When the directional cue was a gaze-averted face, the participants detected the target (dot) more slowly in the invalid condition (385.95 ± 8.32 ms) when compared to the valid condition (377.00 ± 7.82 ms). By contrast, when the directional cue was a head-averted face, the participants performed as quickly on valid trials (388.68 ± 10.22 ms) as on invalid trials (385.44 ± 9.20 ms) (see Figure 3B).

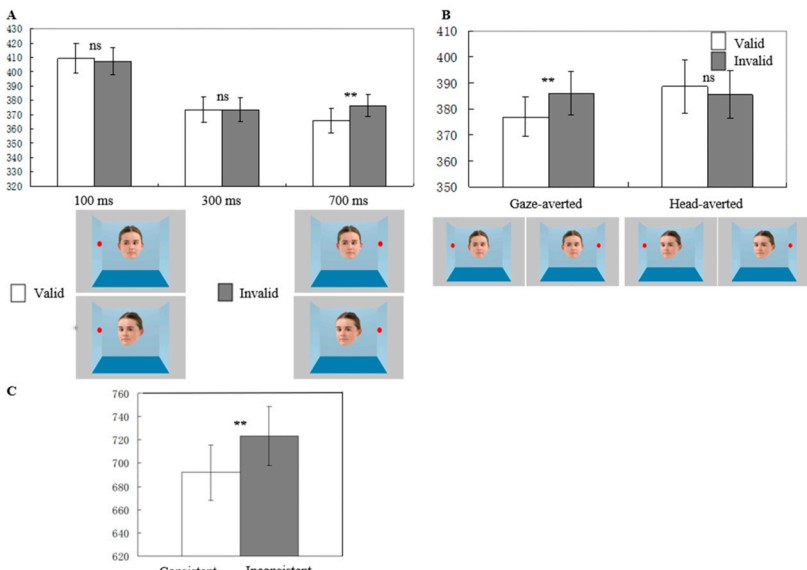

**Figure 3.** Reaction times in each condition of Experiments 1A (**A**,**B**) and 1B (**C**). Means and standard errors are shown; error bars refer to ±1 standard error of mean; ** $p$ < 0.01, ns indicated non-significant difference.

In support of the predictions, there was a stronger cue-validity effect for gaze-averted faces than for head-averted faces. Consistent with Qian et al.'s [17] conclusion, our finding demonstrates that attentional orienting is influenced by perceived gaze with reference to head orientation. Moreover, the cue-validity effect appeared only in a longer SOA of 700 ms, which is at the late temporal stage when volitional/voluntary attention happens [24–27], which fits with orienting effects in previous investigations examining avatars as spatial cues [2,28,29].

### 2.2. Experiment 1B (Dot-Perspective Task)

Experiment 1B examined whether the manipulation of eye–head cues (i.e., gaze-averted face vs. head-averted face) could moderate the consistency effect in the dot-perspective task. The implicit mentalising account would expect comparable effects for both gaze-averted and head-averted faces as visible eye cues persisted. By contrast, the submentalising account would expect weaker effects for head-averted faces than for gaze-averted faces because the modulation effect of the eye-head directional cue on attentional orienting could be generalised to the dot-perspective task.

#### 2.2.1. Participants

A new sample of 42 undergraduates from the IPRP system participated in the experiment. Eight participants were excluded (see 'Results'), leaving 34 volunteers (27 females, mean age: 19.6 years; age range: 18–31 years) for further analysis. Consistent with Fan et al.'s study [12], we used G*power 3.1 to run the priori power analysis for the one within-subject factor 'Consistency'. The analysis recommended that 24 participants were needed to detect the consistency effect found by Samson et al. [1] with an effect size of 0.61, an alpha level of 0.05, and a power of 80%.

#### 2.2.2. Stimuli

All the stimuli were the same as those of Experiment 1A with the exception of the distribution of disc(s) (about $0.7° \times 0.9°$, 3.1–3.2° from the face) that was/were displayed vertically on one or two lateral walls (see examples in Figure 1C,D). Ten dot layouts (that were the same as dot layouts of Samson et al.'s study [1]) were equivalently combined with a gaze-averted or head-averted face. In 50% of trials, the number of disc(s) that the participants could see was the same as the number where the central face was directed (consistent condition), but the participants could see more disc(s) in the remaining trials (inconsistent condition) (see supplemental Appendix B for evaluations of the stimuli).

#### 2.2.3. Procedure

Participants completed the dot-perspective task as illustrated in Figure 2B. Trials began with a fixation cross presented for 750 ms in a virtual room. A 500 ms blank later, the word 'you' (i.e., the participants were instructed to judge self-perspective and ignore the centrally presented faces in the room) was presented for 750 ms. After a 500 ms interval, a digit (0–3) appeared for 750 ms. Then, 500 ms later, the scene of a room with a central stimulus (i.e., the agent's face) and disc(s) on the lateral wall(s) was displayed for a maximum of 3000 ms. The participants were asked to judge whether the scenario of the room that they saw matched the given perspective (i.e., word and number) as quickly and accurately as possible. Specifically, pressing '1' for a 'Yes' response (i.e., matching response) by the forefinger of the left hand, pressing '2' for a 'No' response (i.e., mismatching response) by the forefinger of the right hand. There was a 500 ms interval between trials with the room remaining present.

Every block consisted of 48 test trials (12 consistent with gaze-averted face, 12 inconsistent with gaze-averted face, 12 consistent with head-averted face, 12 inconsistent with head-averted face) and 4 filler trials equally organised into matching and mismatching trials. In filler trials, a face was centrally presented in a room without discs on the wall. Together, the experiment commenced with a block of 26 practice trials followed by 4 blocks of 52 test trials. The trial presentation order was pseudo-randomised in order not to have more than three consecutive trials of the same condition throughout the experiment, and the order was fixed across participants. Trials were arranged into two consecutive blocks for gaze-averted faces and two consecutive blocks for head-averted faces, with the presentation order of blocks counterbalanced across participants.

2.2.4. Results and Discussion

Response time and accuracy were submitted to a 2 × 2 repeated-measures ANOVA with 'Central stimulus type' (Gaze aversion vs. Head aversion) and 'Consistency' (Consistent vs. Inconsistent) as within-subject independent variables. In accordance with Samson et al.'s study [1], only matching trials were analysed. Filler trials were eliminated from further analysis. The dot-perspective task in Experiment 1B (and also in Experiment 2B and Experiment 3) applied the following criteria for data analysis. First, participants with an accuracy of less than 70% on average or 60% in any experimental condition would be excluded from the data set. Furthermore, participants with response times exceeding the range 'mean ± 2.5 SDs' would also be eliminated for further analysis. Experiment 1B eliminated 6 participants due to low accuracy and another 2 participants because of excessively slow response times, leaving data from 34 participants for further analysis. In terms of accuracy, we observed a high accuracy (more than 90% in all experimental conditions) in the dot-perspective task.

Only accurate trials were chosen for analysing response time. Data analysis showed a significant main effect of Consistency ($F$ (1,33) = 9.40, $p$ = 0.004, $\eta_p^2$ = 0.22). Post hoc testing indicated slower response times for the inconsistent trials (mean ± standard error, 723.28 ± 25.27 ms) than for the consistent trials (691.71 ± 23.47 ms) ($t$(33) = −3.07, $p$ = 0.004) (see Figure 3C). No other effects were significant ($ps$ > 0.15).

A Bayesian approach was adopted to further explore the absence of an interaction between Consistency and Central stimulus type. Bayesian statistics can indicate the likelihood for the null hypothesis (i.e., no interaction) to be correct, the alternative hypothesis (i.e., interaction) to be correct, or there is not enough evidence to favour either hypothesis. Bayesian factor ($BF_{01}$) values that are above 3 support the null hypothesis, whereas $BF_{01}$ values that are below 1/3 support the alternative hypothesis. Using JASP to conduct Bayesian statistics [28], the $BF_{01}$ calculated for the Consistency × Central stimulus type interaction was 53.90, favouring the null hypothesis. The posterior probability $P(M | data)$ for the null hypothesis and the alternative hypothesis was 0.99 and 0.01, respectively, which provided strong evidence to support the null hypothesis.

We found comparable consistency effects for both gaze-averted and head-averted faces, supporting predictions based on the implicit mentalising account. Our findings extend previous findings of implicit mentalising triggered by a virtual avatar Samson et al. [1] to a real human's face. Similarly, in our experiment, the existence of implicit mentalising was reflected by the altercentric interference effects for both kinds of faces—participants' relatively automatic calculation of the agent's visual perspective interfered with their own perspectives, and thereby, they judged their own perspectives more slowly when the two types of perspectives were different compared with when they were the same. More intriguingly, unlike the modulation of attentional orienting by eye-head cues in Experiment 1A, the eye-head cues' manipulation was not sufficient to modulate visual perspective-taking in Experiment 1B. Instead, the implicit mentalising account and previous studies [11,12] show that the agent's visual access may be crucial for affecting implicit mentalising in the dot-perspective task. Because both gaze-averted and head-averted faces have visible averted-gaze, the two kinds of faces trigger implicit mentalising. In sum, the findings in Experiments 1A and 1B favour the contribution of implicit mentalising to the consistency effect in the dot-perspective task.

Apart from eye and head directions, finger-pointing direction is another effective and important cue in social interactions. Visual access to current eye–head cues predominate the implicit mentalising in the dot-perspective task. Experiments 2 and 3 explored whether the influence of visual access on implicit mentalising persisted when eyes were presented simultaneously with finger-pointing, which might provide new insight into the implicit mentalising vs. submentalising debate.

## 3. Experiment 2

The supportive evidence of Experiment 1 for the existence of implicit mentalising via using eye–head cues provided the impetus to find a new effective way to clarify the debate, namely, manipulating the agent's different body parts and comparing their influences on cue-validity effect in the Posner task with those on consistency effect in the dot-perspective task.

As described above, the agent's finger-pointing showed the superiority of capturing others' attention relative to the agent's eye gaze for young children in the modified Posner Task [23]. However, whether the superiority applies to adults in a classic attentional-orienting task (i.e., Posner task) remained unknown. Experiment 2A addressed this issue by manipulating eye–finger-pointing cues (gaze-averted-agent vs. finger-pointing agent) and comparing the cues' effects on attentional orienting in the Posner task. Furthermore, the role of finger-pointing in L1VPT processing and its comparison with the role of eye gaze remained unknown. Experiment 2B measured the influence of finger-pointing on the consistency effect of the dot-perspective task and compared it with the influence of eye gaze. Altogether, Experiment 2 attempted to explore whether manipulation of eye–finger-pointing cues could dissociate attentional orienting from implicit mentalising as eye–head cues did.

### 3.1. Experiment 2A (Posner Task)

Experiment 2A explored whether eye–finger-pointing cue manipulation (i.e., gaze-averted agent vs. finger-pointing agent) would moderate adults' attentional orienting via the cue-validity effect in the Posner task. Dovetailing with Gregory et al.'s findings [23], we posited that compared with gaze-averted agents, finger-pointing agents should evoke stronger cue-validity effects in the Posner task. Our prediction was based on the claim that finger-pointing is a stronger cue than gaze-averted for adults. The claim was raised on the basis of the following two findings. First, the former one (i.e., three dimensions) is more perceptually salient than the latter one (i.e., two dimensions) [30]. Second, the dimension with greater perceptual variation has been found to elicit a stronger interference effect on the categorisation of the dimension with weaker perceptual variation than vice versa [30].

#### 3.1.1. Participants

A new group of 38 undergraduates signed up to the IPRP system for experimental participation with three participant exclusions (see 'Results'). The sample size of 35 participants (28 females, mean age: 19.3 years; age range: 18–30 years) was chosen based on the priori power analysis in Experiment 1A.

#### 3.1.2. Stimuli and Procedure

Photographs of the volunteer included her face and upper part of the body. Separating eye direction from the body gesture's direction, the gaze-averted person (i.e., a person with 45° left or right averted eyes and with no body gesture) and the finger-pointing person (i.e., a person with direct eye-gaze and approximately 45° left or right averted finger-pointing) were generated. The gaze-averted person could see the dot(s) where her eyes were directed, whereas the finger-pointing person could not see the dot(s). In fitting with the central stimuli, the sizes of all the stimuli in the scene were changed compared with those of the stimuli in Experiment 1. Specifically, the participant (viewing at a distance of about 72 cm from a 14-inch computer screen) would observe a scene where a gazer was centrally presented and a disc (about $1.0° \times 1.2°$, $4.4°$ from the face) was arranged on one lateral wall ($14.3° \times 13.3°$). An image of the gazer consisted of a frontal-view face ($2.9° \times 3.8°$) with $0.2°$-width-eyes and a frontal-view upper body ($7.4° \times 3.8°$, see examples in Figure 4A,B) (see supplemental Appendix C for evaluations of the stimuli). The scenarios of valid and invalid conditions were similar to those of Experiment 1A. The current procedure was the same as Experiment 1A's procedure.

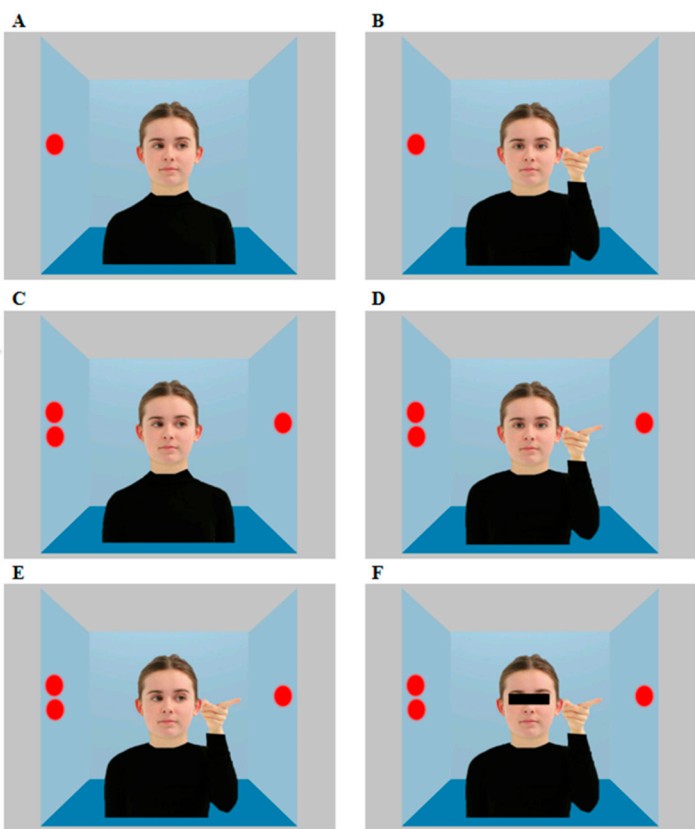

**Figure 4.** Examples of room images used in Experiment 2A ((**A**): Gaze-averted Agent, (**B**): Finger-pointing Agent), Experiment 2B ((**C**): Gaze-averted Agent, (**D**): Finger-pointing Agent), and Experiment 3 ((**E**): Eyes-opened–finger-pointing Agent, (**F**): Eyes-covered–finger-pointing Agent).

### 3.1.3. Results and Discussion

The methods for data analysis, as well as the standards for data exclusion, were the same as those of Experiment 1A. Three participants were excluded from the data analysis because their response times were not within the range 'mean $\pm$ 2.5 SDs', leaving 35 participants' data for analysis. The 3 (SOA: 100 ms vs. 300 ms vs. 700 ms) $\times$ 2 (Central stimulus type: gaze-averted vs. finger-pointing) $\times$ 2 (Validity: valid vs. invalid) repeated-measures ANOVA was carried out for data analysis, and the percentage of error and response time were adopted as dependent variables. On average, the participants made erroneous responses in 5.67% of filler trials.

With regard to reaction time, only correct trials of no-filler trials were analysed. We found a significant SOA $\times$ Validity interaction effect ($F$ (2,68) = 6.03, $p$ = 0.004, $\eta_p^2$ = 0.15). Paired-sample t-tests showed a significant Validity effect when SOA duration was 700 ms ($t$(34) = −4.44, $p$ < 0.001) but not 100 ms ($t$(34) = −0.52, $p$ = 0.61) or 300 ms ($t$(34) = −1.51, $p$ = 0.14). Specifically, when the SOA duration was 700 ms, the participant detected the targeted dot more slowly in the invalid condition (355.09 $\pm$ 8.03 ms) relative to the valid condition (340.94 $\pm$ 7.25 ms). In contrast, the participant completed the Posner task as quickly in the valid condition as in the invalid condition, both when the SOA duration was 100 ms and when the SOA duration was 300 ms (see Figure 5A). Furthermore, we found a marginally significant interaction effect between the Central stimulus type and Validity ($F$(1,34) = 3.45, $p$ = 0.072, $\eta_p^2$ = 0.092). Specifically, we observed a slower response time in the Posner task for the invalid trials (368.83 $\pm$ 8.03 ms) than for the valid trials (359.03 $\pm$ 7.46 ms) when the directional cue was a finger-pointing agent. In contrast, we did not find a significant difference in response time between the valid condition (362.34 $\pm$ 7.56 ms) and the invalid condition (365.39 $\pm$ 7.44 ms) when the directional cue was a gaze-averted agent (see Figure 5B). However, the three-way interaction effect was not

significant ($F$ (2,68) = 0.79, $p$ = 0.45, $\eta_p^2$ = 0.023), indicating the cue-validity effect restricted to longer SOAs is driven by both averted gaze and finger-pointing.

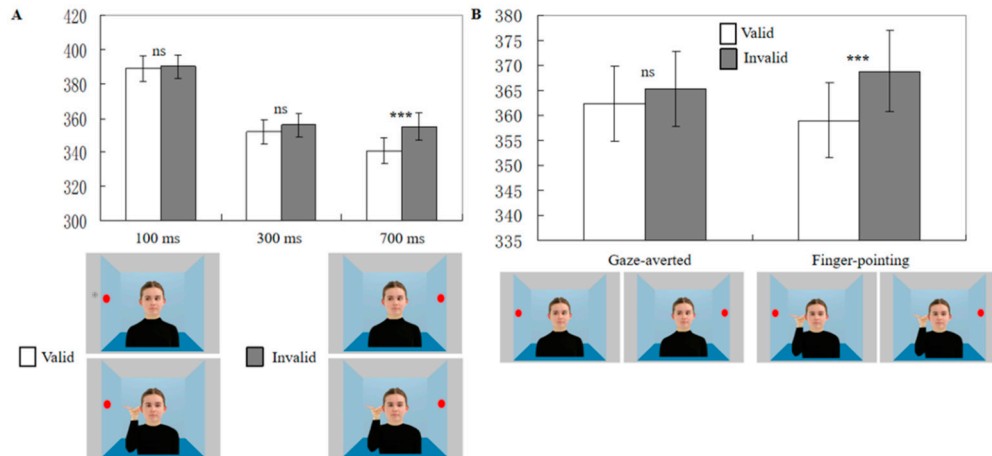

**Figure 5.** Reaction times in each condition of Experiment 2A (**A**,**B**). Means and standard errors are shown; error bars refer to ±1 standard error of mean; *** $p$ < 0.001, ns indicated non-significant difference.

Consistent with Experiment 1A's, there was a volitional attentional orienting effect (i.e., a cue-validity effect occurred only at a longer SOA of 700 ms) that was triggered by a directional cue (i.e., finger-pointing direction). More intriguingly, there was a greater cue-validity effect induced by finger-pointing relative to gaze-averted, supporting our expectations. The resulting pattern was consistent with previous findings of preschool children showing superior performance for lateral-oriented finger-pointing than for gaze-averted [23]. However, the non-significant cue-validity effect for gaze-averted agents is inconsistent with the classic gaze cueing effect. One possible interpretation may be the salience of the directional information. Specifically, an averted gaze may not be sufficient to trigger attentional orienting when compared to finger-pointing direction, as finger-pointing is relatively salient, so an averted gaze may be ignored.

### 3.2. Experiment 2B (Dot-Perspective Task)

Experiment 2B examined whether implicit mentalising could be generated in the dot-perspective task by manipulating eye–finger-pointing cues (i.e., gaze-averted agent vs. finger-pointing agent). We proposed two predictions based on the two competing accounts. The implicit mentalising account emphasising the importance of line of sight would predict a consistent effect of the dot-perspective task for gaze-averted agents as visual access was available. Heyes's [3] submentalising claim that attentional orienting contributes to the consistency effect of the dot-perspective task would lead to a prediction consistent with the findings of the Posner task (a classic task tapping attentional orienting) in Experiment 2A. Specifically, the effect in the dot-perspective task would be present for finger-pointing agents but not for gaze-averted agents (given that a finger-pointing cue is more perceptually salient than a gaze-aversion cue).

#### 3.2.1. Participants

Another 52 undergraduates from the IPRP system took part in the experiment, and 10 participants were excluded from the data set (see 'Results'). Based on the power analysis of Experiment 1B, the sample size of 42 (34 females, mean age: 19.9 years; age range: 18–42 years) exceeded the 24 participants required to detect Samson et al.'s consistency effect [1].

#### 3.2.2. Stimuli and Procedure

All the stimuli were the same as those of Experiment 2A with the exception that the disc(s) (about 1.0° × 1.2°, 4.4–4.6° from the face) were vertically presented on one or two

lateral walls (see examples in Figure 4C,D) (see supplemental Appendix C for evaluations of the stimuli). The disc distributions accorded with Experiment 1B's distributions. The number of disc(s) that the participant could see and the agent was directed to was the same in the consistent condition but were different in the inconsistent condition. The current procedure was identical to Experiment 1B's procedure.

3.2.3. Results and Discussion

The methods for data analysis and data exclusion criteria were identical to Experiment 1B.

Nine participants were excluded due to low accuracy, and another participant was eliminated as their response time exceeded the range 'mean + 2.5 SDs'. Thus, 42 participants' data were selected for the 2 (Central stimulus type: Gaze-averted vs. Finger-pointing) × 2 (Consistency: Consistent vs. Inconsistent) repeated-measures ANOVA, and accuracy and reaction time were dependent variables. We observed a high accuracy (more than 90% in all experimental conditions) in the dot-perspective task.

For reaction time, only correct trials were selected for data analysis. We found a significant Central stimulus type × Consistency interaction effect ($F$ (1,41) = 6.41, $p$ = 0.015, $\eta_p^2$ = 0.14). Paired-sample t-tests revealed a Consistency effect when the central stimulus was a finger-pointing agent ($t$(41) = −5.22, $p$ < 0.001) but not a gaze-averted agent ($t$(41) = −1.06, $p$ = 0.29). Specifically, the finger-pointing agent made participants judge their own perspective more slowly in the inconsistent condition (763.74 ± 28.05 ms) compared to the consistent condition (717.89 ± 25.21 ms). In contrast, the gaze-averted agent made participants judge their own perspective as fast in the consistent condition (719.06 ± 25.23 ms) as in the inconsistent condition (731.04 ± 28.17 ms) (see Figure 6A).

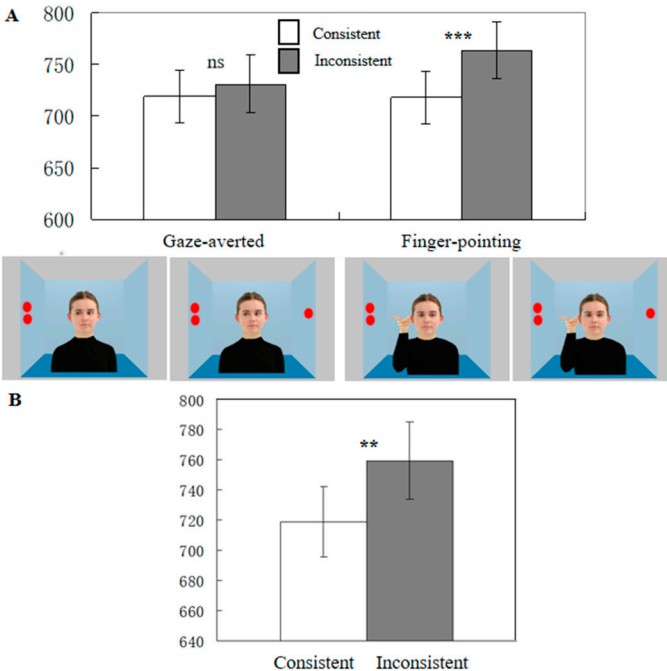

**Figure 6.** Reaction times in each condition of Experiments 2B (**A**) and 3 (**B**). Means and standard errors are shown; error bars refer to ±1 standard error of mean; ** $p$ < 0.01, *** $p$ < 0.001, ns indicated non-significant difference.

Fitting with the submentalising hypothesis, there was an effect triggered by finger-pointing agents but not gaze-averted agents in both Posner and dot-perspective tasks. One possible interpretation is that the superiority effect of finger-pointing in the Posner task of Experiment 2A may be generalised to the dot-perspective task of Experiment 2B. Instead of averted gaze, finger-pointing may predominate in generating the effect

in the dot-perspective task via an attentional orienting mechanism. However, this is the first experiment exploring the debate by manipulating eye–finger-pointing cues and comparing the effects in both Posner and dot-perspective tasks, which needs to be further unpacked. Furthermore, finger-pointing may be so salient that the role of averted gaze is decreased to be non-significant in both Posner and dot-perspective tasks. Thus, Experiment 3 manipulated the agent's visual access (a widely used manipulation to clarify the debate) and combined it with finger-pointing manipulation to further address the issue.

## 4. Experiment 3

Experiment 3 investigated the implicit mentalising vs. submentalising debate by combining visual access manipulation (i.e., eyes-opened vs. eyes-covered) with finger-pointing in the dot-perspective task. Both Fan et al. [12] and Furlanetto et al. [11] manipulated visual access and found a consistency effect in the visible condition but not the invisible condition. The findings emphasized the importance of visual access and demonstrated people's capability to automatically calculate the agent's visual perspective, supporting the implicit mentalising account. However, in Experiment 2, finger-pointing showed the superiority of capturing others' attention relative to averted gaze not only in the Posner task (tapping attentional orienting) but also in the dot-perspective task (tapping L1VPT). The findings raised a question about whether visual access could still predominate the processes in the dot-perspective task when finger-pointing needed to be considered. Thus, to address the issue, manipulation of the agent's line of sight and finger-pointing was combined, and their influences were compared on the dot-perspective task. Based on Fan et al.'s findings [12] and the implicit mentalising account emphasising the importance of visual access, one prediction was the significant consistency effect for eyes-sighted agents but not for eyes-covered agents and its stability when finger-pointing's directional information was simultaneously presented. According to Experiment 2's findings and the submentalising account emphasising the importance of attentional orienting, another prediction was comparable effects for both eye-opened finger-pointing and eye-covered–finger-pointing agents.

### 4.1. Participants

Fifty-seven undergraduates from the IPRP system were recruited for participation, excluding 16 participants (see 'Results'), leaving 41 participants (33 females, mean age: 19.9 years; age range: 18–33 years) for further analysis. Based on the power analysis in Experiment 1B, the sample size exceeded the 24 participants needed to detect Samson et al.'s consistency effect [1].

### 4.2. Stimuli and Procedure

Photographs of the volunteer with averted gaze, as well as finger-pointing (i.e., a person with approximately 45° left or right averted finger-pointing that was compatible with eye direction), were taken for Experiment 3. The pictures consisted of two seeing conditions: eyes-opened–finger-pointing (i.e., a finger-pointing agent with 45° left/right-averted eye-gaze) vs. eyes-covered–finger-pointing (i.e., a finger-pointing agent with a black rectangle on her eyes, 2.2° × 0.6°) (see examples in Figure 4E,F). The size of all the stimuli and the disc distributions were the same as those of Experiment 2B. Before the formal experiment, all the participants were asked to determine whether the agent could see the three discs on a lateral wall where the agent was directed. All of them reported that the eyes-opened person could see (visible condition), whereas the eyes-covered person could not (invisible condition). The procedure of Experiment 3 was the same as that of Experiment 2B except for the evaluation of the visibility of the central stimuli.

### 4.3. Results and Discussion

The methods for data analysis, as well as the standards for data exclusion, were the same as those of Experiment 1B. A two-way repeated-measures ANOVA with Central stim-

ulus type (Eyes-opened–finger-pointing vs. Eyes-covered–finger-pointing) and Consistency (Consistent vs. Inconsistent) as independent variables, and accuracy and response time as dependent variables was performed. Sixteen participants were excluded from further analysis, including fourteen participant eliminations because of low accuracy and another two participant exclusions due to excessively slow response times. We found that accuracy was higher than 90% in all experimental conditions.

With respect to response time, only accurate trials were selected for data analysis. ANOVA analysis only showed a significant main effect of Consistency ($F$ (1,40) = 13.33, $p$ = 0.001, $\eta_p^2$ = 0.25). Post hoc testing indicated that the participants judged their own perspective more slowly in the inconsistent condition (759.55 $\pm$ 25.55 ms) than in the consistent condition (718.75 $\pm$ 23.24 ms) ($t$(40) = −3.65, $p$ = 0.001) (see Figure 6B). Other effects failed to reach significance (all $p$s > 0.05). A Bayesian approach was employed to further explore the absence of a Central stimulus type $\times$ Consistency. Computed using JASP, the $BF_{01}$ for the interaction was 44.72, which supported the null hypothesis. The posterior probability $P(M \mid \text{data})$ for the null hypothesis was 0.99, whereas for the alternative hypothesis, it was 0.01, providing strong evidence to favour the null hypothesis.

There was a comparable magnitude of consistency effects for both eyes-sighted–finger-pointing and eyes-covered–finger-pointing agents, favouring the hypothesis based on the submentalising account. The finding did not replicate Fan et al.'s [12] findings of the elicitation of consistency effect in the visible condition but not in the invisible condition, even though these two experiments used the same manipulation of visual access. The inconsistent findings may be due to different roles and interactions between different body cues. In Fan et al.'s [12] Experiment 1, for eye–head cues, visual access instead of head direction may predominate the processing in the dot-perspective task. In contrast, the mechanism of the dot-perspective task may be changed when finger-pointing is involved. The lack of modulation of visual access in the dot-perspective task of Experiment 3, when finger-pointing was simultaneously presented, suggests that finger-pointing may override visual access to predominate the processing in the dot-perspective task via an attentional orienting mechanism. This fits with the possible explanation of Experiment 2's findings. Taken together, the findings of Experiments 2 and 3 support the contribution of the finger-pointing-triggered attentional orienting to the effect in the dot-perspective task.

## 5. General Discussion and Conclusions

We carried out three experiments in the study, attempting to cast light on the implicit mentalising vs. submentalising debate. Overall, Experiment 1 discovered that manipulation of eye–head cues (gaze-averted face vs. head-averted face) could dissociate attentional orienting from implicit mentalising to support the contribution of implicit mentalising to the dot-perspective task's consistency effect. However, the attentional orienting mechanism drove the dot-perspective task when finger-pointing was combined with eye-related manipulations in Experiments 2 and 3.

1. **Eye–head cues modulated attentional orienting but not implicit mentalising**

Experiment 1A found a significant cue-validity effect for gaze-averted faces, which was consistent with both our hypothesis and previous findings showing attentional orienting towards a real human's face with an averted gaze [17,31,32].

More importantly, consistent with Qian et al.'s [17] work, we found a greater cue-validity effect for gaze-averted than for head-averted faces, demonstrating that attentional orienting can be modulated by an eye–head directional cue in the Posner task. Unlike Qian et al.'s [17] finding of a weaker effect for head-averted faces, we did find a non-significant effect for head-averted faces. The discrepancy may be because of a bigger difference in gaze angle evaluation between gaze-averted and head-averted faces of the present experiment (i.e., 2.6) compared with Qian et al.'s [17] study (i.e., 0.8). In the present study, the greater difference may increase the salience of gaze-averted faces and decrease the salience of head-averted faces, which raises the possibility of the non-significant cue-validity

effect for head-averted faces. Nevertheless, we are mindful that cross-study comparisons are difficult to provide definitive evidence (e.g., lack of randomisation to conditions).

Experiment 1B found that when the widely used virtual avatar was substituted with a real human's face to be the central stimulus, the consistency effect was robustly yielded. Moreover, we observed a comparable magnitude of consistency effects for both gaze-averted and head-averted faces in the dot-perspective task. Even though the agent's eyes of head-averted faces were not absolutely directed to the dot(s) on the wall, the agent's visual access was available; ratings of the gaze direction of head-averted faces were 1.6 in Experiment 1 (please see Appendix A). As both gaze-averted and head-averted faces have visible averted gaze, the two kinds of faces trigger implicit mentalising. An intuitively appealing account of elicitation of the consistency effect in both types of faces is that the always available visual access instead of the eye–head directional information may allow the participants' relatively automatic computation of the agent's visual perspective, thereby yielding the consistency effect for both kinds of faces under Self-perspective instruction. Visual access may play a crucial role in triggering implicit mentalising for processing eye–head cues in the dot-perspective task. These findings, to a certain extent, lend support to the existence of implicit mentalising in the dot-perspective task.

2. **Finger-pointing of eye–finger-pointing cues may predominate in generating the effect in the dot-perspective task via an attentional-orienting mechanism**

Compared with eye–head cues, eye–finger-pointing cues show a different mechanism in processes of both the Posner task and dot-perspective task, providing a new viewpoint to understand the implicit mentalising vs. submentalising debate.

Experiment 2A supports our hypothesis by finding a greater cue-validity effect triggered by finger-pointing agents than by gaze-averted agents in the Posner task. The superiority of finger-pointing extended Gregory et al.'s [23] findings, showing young children's better performance for finger-pointing relative to eye gaze in a modified Posner task by revealing the superiority in adults as well as in the classic Posner task. One possible explanation for the findings is that the greater perceptual variation of finger-pointing (i.e., three dimensions) than that of averted gaze (i.e., two dimensions) can make finger-pointing cues more recognisable and stronger as a directional cue, triggering greater attentional orienting compared with gaze-averted cues. The interpretation also fits with Butterworth's statement [20] that finger-pointing direction may be a more accurate spatial cue relative to eye direction.

Experiment 2B associated eye–finger-pointing cues with visual perspective-taking via the consistency effect of the dot-perspective task. We found a greater consistency effect for finger-pointing agents than for gaze-averted agents, supporting our submentalising hypothesis. The same resulting pattern of Experiments 2A and 2B may reveal the same mechanism for processing both the Posner and the dot-perspective task: finger-pointing direction rather than gaze direction triggers volitional attentional orienting. Different from stressing the importance of visual access in eye–head cues of the dot-perspective task, finger-pointing may predominate in generating the effect in the dot-perspective task via an attentional orienting mechanism for the eye–finger-pointing cues. Overall, Experiment 2's findings favour the contribution of the finger-pointing elicited attentional orienting to the consistency effect in the dot-perspective task.

However, it is a novel experiment exploring the effect of finger-pointing in both the Posner task and the dot-perspective task and comparing the effects with those triggered by averted gaze. Further, it is possible to question whether the manipulation of eye gaze in Experiment 2 was salient enough to trigger the potential effects when compared with finger-pointing. Experiment 3 adopted an apparent, easily recognisable gaze-related manipulation (i.e., manipulating the agent's visual access to create visible and invisible conditions) as the manipulation has been found to modulate the consistency effect in the dot-perspective task. Furthermore, we compared the manipulation of visual access with that of finger-pointing in modulating the effect in the dot-perspective task.

Experiment 3 observed a comparable magnitude of consistency effects for both eyes-sighted–finger-pointing and eyes-covered–finger-pointing agents, supporting our submentalising hypothesis. The findings fit with neither Furlanetto et al.'s [11] nor Fan et al.'s [12] findings of a significant consistency effect in the visible condition but not in the invisible condition. The contradictory findings may be because the processing mechanism of the dot-perspective task is changed when considering finger-pointing. In the two previous studies, manipulation of the agent's visual access alongside beliefs of clearly distinguishing visible conditions from invisible conditions can modulate the consistency effect. The effect was absent even though directional cue(s) (i.e., head + torso directions or only head direction) are identical for both the seeing and non-seeing conditions. In these two studies, visual access is superior to directional cueing (head + torso/head directions) in modulating implicit mentalising in the dot-perspective task, which supports the implicit mentalising account. However, in the current experiment, the dot-perspective task's effect induced by finger-pointing direction cannot be modulated by the manipulation of visual access. The line of sight manipulation was the same as Fan et al.'s [12] manipulation. Nevertheless, a comparable magnitude of effects for the two visibility conditions was observed. Thus, finger-pointing direction may be superior to visual access in moderating the effect in the dot-perspective task when finger-pointing is considered, supporting that the finger-pointing-generated attentional orienting contributes to the dot-perspective task's consistency effect. The interpretation also fits with the attentional orienting mechanism for processing Experiment 2B's dot-perspective task.

Taken together, Experiment 1's findings support the existence of implicit mentalising; the findings of Experiments 2 and 3 together favour the contribution of attentional orienting to the consistency effect in the dot-perspective task. The findings of our three experiments suggest an integration between implicit mentalising and submentalising processes in the dot-perspective task. We did not speak against the contribution of attentional orienting to the consistency effect in the dot-perspective task. Instead, we thought that the generation of the effect was related to mentalising even though attentional orienting existed in the dot-perspective task. The reasons are as follows. First, Fan et al.'s findings [12] suggested that mentalistic processing of seeing (i.e., the belief of the agent being seeing) elicited the consistency effect in the dot-perspective task. Additionally, Bukowski et al.'s findings [2] indicated that it was not the presence of the avatar (with the directional cue) itself but the perspective-taking task instruction that triggered the implicit computation of what someone else was looking at by prioritising the attention towards another person. Accordingly, the integration between the implicit-mentalistic and the sub-mentalistic processes can explain the consistency effect in the dot-perspective task. Additionally, visual access is more salient than head orientation when considering gaze and head directions, whereas visual access is less salient than finger-pointing direction when considering gaze and pointing directions. Therefore, the hierarchy in the salience of the cues may explain why visual-access-generated implicit mentalising dominates in processing eye–head cues and finger-pointing-generated attentional orienting dominates in processing eye–finger-pointing cues in the dot-perspective task. This is compatible with the statements that neither implicit mentalising nor submentalising accounts of the dot-perspective task are fully correct [33] and that an integrated framework combining the attribution of mentalising and the operation of submentalising processes can account for social orienting [34].

Although finger-pointing appears not to trigger implicit mentalising in the dot-perspective task, this cannot rule out the possibility that finger-pointing may generate mentalising-related processes. Tomasello and Camaioni [35] claimed that using finger-pointing for declarative purposes depends on a person's mentalising ability, and further, the production of declarative finger-pointing has been found to be positively correlated with the comprehension of intentions [36]. Consequently, further work is needed to examine whether finger-pointing can trigger mentalising-related processes and to compare the contribution of finger-pointing with that of visual access cues on other mentalising tasks. Additionally, because the participants in the current study were all adults, whether the

effects can be modulated by age (i.e., children, adolescents, adults and older individuals) needs further investigation. Furthermore, as high-functioning Autism Spectrum Disorder (ASD) individuals did not succeed in implicit mentalising [37], clarifying the processing mechanism of the implicit mentalising vs. submentalising debate may be beneficial for deeply understanding ASD's deficit in implicit mentalising, which warrants further exploration. Additionally, since attention has been found to be correlated with blood carbon dioxide levels [38], physiological parameters may be adopted to measure attention and submentalising processes when the implicit mentalising vs. submentalising debate is investigated.

In conclusion, rather than contrasting the two accounts, our work offers an integration between implicit-mentalistic and sub-mentalistic processes in processing the dot-perspective task. When eye–head cues are processed, visual access is superior to the head direction in driving implicit mentalising. However, when eye–finger-pointing cues are processed, the role of visual access may be interfered with by finger-pointing, which can predominate the effect of the dot-perspective task via an attentional orienting mechanism. Overall, our work opens up a new possibility where implicit mentalising and submentalising accounts are not mutually exclusive, yielding a deeper understanding of the human capacity for visual perspective-taking.

**Author Contributions:** C.F. conceptualised and designed the online experiments, conducted the research, analysed the data and wrote the manuscript. T.S. and J.L. helped with conceptualising the experiments and with the writing and editing of the manuscript. All authors have read and agreed to the published version of the manuscript.

**Funding:** This research received no funding.

**Institutional Review Board Statement:** The study protocol was approved by the School of Psychology Human Ethics Committee under the delegated authority of Victoria University of Wellington's Human Ethics Committee. This study followed the Declaration of Helsinki. All methods were carried out in accordance with relevant guidelines and regulations.

**Informed Consent Statement:** Written-informed consent was obtained from all subjects (including those whose images were used in the study experiments) for publication of identifying information/images in an online open-access publication.

**Data Availability Statement:** The data and materials are available upon reasonable request to the first author.

**Acknowledgments:** The authors thank Lin Hu and Xinglong Yao for their help with uploading the experimental procedures online. Thanks, Lisa Woods, Geraldine Smieszhala and Mingming Zhang, for their help with power analysis. The authors also thank Yanzhu Chen and Mingjie Xu for their help with the photographs taken.

**Conflicts of Interest:** The authors declare no conflict of interest.

## Appendix A. Manipulation Check of Eye–Head Directional Cue in Experiment 1A

### A.1. Visibility Evaluation

Before Experiment 1A's Posner task, 20 adults (who would not participate in the formal experiment) evaluated whether the centrally presented face could see the dot in valid or invalid conditions. Eight images involving one dot on the left/right wall and a face with left/right-viewing eye-gaze (i.e., gaze-averted face) or with a left/right-oriented head (i.e., head-averted face) were displayed. All the participants reported that the gaze-averted but not head-averted faces could see the dot in the valid condition, whereas neither of the two kinds of faces could see the dot in the invalid condition.

### A.2. Gaze Angle Estimation

In addition, a new sample of 22 adults without participation in the formal experiment evaluated the gaze angle of the four faces shown in Table A1. The gaze angle was rated

from 0 (i.e., direct gaze) to 5 (i.e., extremely left or right gaze). The average ratings for four faces were 2.9 (gaze-averted) and 2.1 (head-averted) when evaluating Qian et al.'s (2013) stimuli, 4.2 (gaze-averted) and 1.6 (head-averted) when evaluating our stimuli. A paired-sample *t*-test showed that the rating of gaze-averted faces was significantly higher than that of head-averted faces in these two studies (all *p*s < 0.05). Among the results, the ratings of Qian et al.'s (2013) stimuli are similar to Qian et al.'s (2013) evaluations (i.e., 2.6 for gaze-averted faces and 1.9 for head-averted faces).

**Table A1.** Stimuli used in Qian et al.'s (2013) and the present studies.

| Type of Faces | Gaze-Averted-Face | Head-Averted-Face |
|---|---|---|
| Qian et al.'s (2013) study | 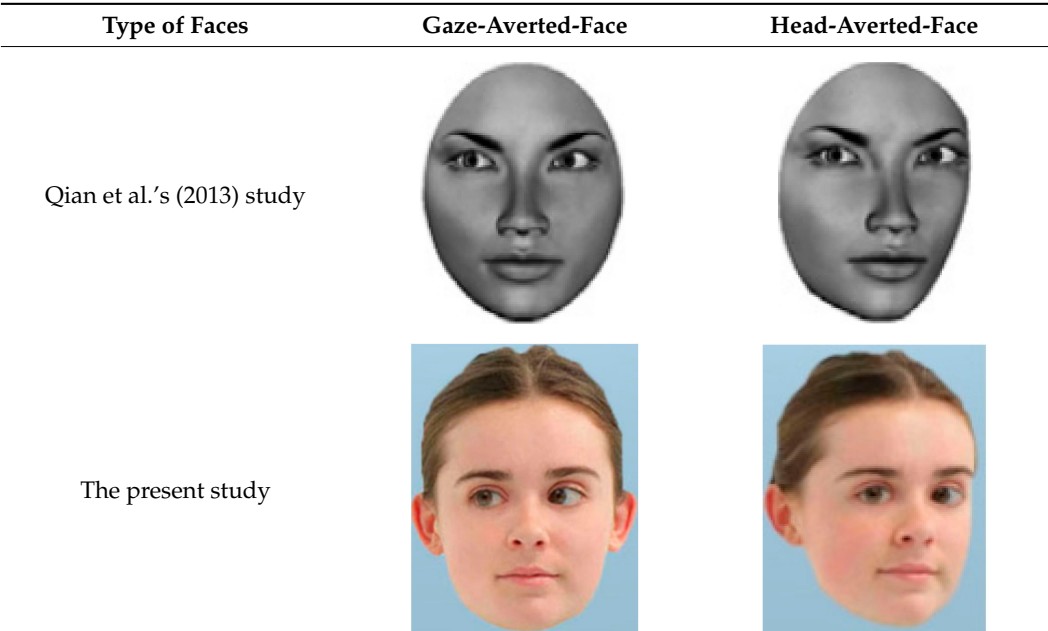 | |
| The present study | | |

Examples of face images used in Qian et al.'s (2013) study (second row) and present study (third row).

## Appendix B. Manipulation Check of Eye–Head Directional Cue in Experiment 1B

*Visibility Evaluation*

The visibility evaluation conducted before Experiment 1B's dot-perspective task was the same as the visibility evaluation in Experiment 1A, except for the vertical presentation of two dots on one lateral wall. Every participant reported that the gaze-averted faces could see the dots, whereas the head-averted faces could not in the consistent condition; neither of them could see the dots in the inconsistent condition.

## Appendix C. Manipulation Check of Eye–Finger-Pointing Cues in Experiment 2

Twenty adults who did not take part in the formal experiment completed a visibility evaluation before the formal experiment. The visibility evaluations in Experiment 2A and 2B were the same as the evaluations in Experiment 1A and 1B, respectively, except for replacing eye-head cues (gaze-averted face vs. head-averted face) with eye–finger-pointing cues (gaze-averted agent vs. finger-pointing agent).

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
