# Peer review of "Unveiling the Dot-Perspective Task: Integrating Implicit-Mentalistic with Sub-Mentalistic Processes"

_psych, doi:10.3390/psych5040078_

Round 1

Reviewer 1 Report

Comments and Suggestions for Authors

Overall, I do like the paper and certainly think it will contribute nicely to the literature. It is very well written and clear. I cannot however recommend publication in its current form. I have two central issues. 1) Interpretation of the data, and 2) contact with the literature.

1. The authors found comparable consistency effects for both gaze-averted and head-averted stimuli in Exp 1B. They then state that this supports the implicit mentalising account. I don’t actually see how this can be. An agent’s perspective is of course dependent on where they are looking, not which direction their head is facing. The authors essential found a consistency effect for a stimulus that cannot see the stimulus (i.e., the head).

2. At the beginning of Line 58, the authors present arguments stating why the Santiesteben et al. study does not rule out the mentalising account. They for instance state that even non-human features can be regarded by participants as having human agency. This and their other arguments are social cognition arguments. The more fundamental problem with the Santiesteben et al. reasoning/rationale was pointed out by Cole, Atkinson, Le, & Smith (2016; given on pages 165-166). The argument is that simply replicating the classic arrow cueing effect cannot tell us very much, if anything, about why the dot-perspective effect works. This argument should be included in the present ms.

The authors state that “One approach that has been widely used to clarify the ongoing debate is to manipulate the avatar’s visual access”. The first study to do this (in the context of mentalising and perspective-taking) was Cole, Smith, & Atkinson (2015). As those authors stated, they took the visual access idea from animal behaviour work. This should be included. Similarly, the present authors state that there are problems with a number of previous experiments that manipulate visual access. For instance, the authors are correct in saying that it is difficult for participants to keep track of what an agent can see (with goggles). This and all the problems outlined are not present in the Cole, Smith, & Atkinson (2015) study, neither in the Cole, Atkinson, Le, & Smith (2016) study. All the experiments in these two papers employ “an easily identifiable barrier”. In fact, one experiment had the participant facing a real person who was sat next to a physical barrier blocking their view of the critical stimuli. These studies should be cited.

Small point. The reference numbering needs checking. For instance, Tomasello, M., & Camaioni, L. is given as 34 in the text but is 35 in the reference section.

Reviewer 2 Report

Comments and Suggestions for Authors

The current study conducted a series of experiments in order to shed more light on the debate about the relationship between implicit mentalizing vs. submentalising processes. The findings of the three experiments suggested an integration between implicit mentalizing and submentalising processes in the dot-perspective task.

The topic is interesting and relevant to the journal’s scope. It introduces an original research question. It has the potential to make a valuable contribution to the research field.

The abstract provides relevant knowledge. However, it is essential to improve its structure to better introduce the reader to the topic and the main objectives of this study.

 The introduction adequately summarizes the current state of the topic. It addresses the limitations of current knowledge in this field.

The methods followed in the experiments are adequately described.

The main text is well-structured and helps the reader follow the procedures.

The general discussion and conclusions provide relevant information. However, the integration of discussion with conclusions is not recommended. The section is too lengthy. It is useful to keep the general discussion as a separate section and add a conclusions section where you can select the most critical comments, based on the results as a whole, highlight the contribution of this research for future research, and provide suggestions for future research. In the discussion section, it is essential to mention some limitations of this research. In addition, in the discussion section, I would suggest taking into consideration the factor of physiological operations on attention and submentalizing processes. I would  suggest the paper: Breathing, attention & consciousness in Sync: The Role of Breathing Training, metacognition & Virtual Reality. Technium Soc. Sci. J.29, 79.

Figures are useful. It is essential to remove additional descriptions and keep the captions. The data is presented clearly and the figures are consistent with the description in the text.

Please make a check in the references. For instance, the study conducted by Fan, Susilo, and Low (2020) is mentioned after the 7th reference but in the reference list is placed in the number 28. In addition, you can check that in the text, all reference numbers are placed in square brackets [ ] (i.e. lines 63, 88). In addition, check whether the reference list should follow the ACS style guide.

Round 2

Reviewer 1 Report

Comments and Suggestions for Authors

I am happy with the revision.